# Management of Parathyroid Disease during the COVID-19 Pandemic

**DOI:** 10.3390/jcm10050920

**Published:** 2021-02-26

**Authors:** Nivaran Aojula, Andrew Ready, Neil Gittoes, Zaki Hassan-Smith

**Affiliations:** 1School of Medicine, Faculty of Medicine, Imperial College, London SW7 2AZ, UK; nivaran.aojula16@imperial.ac.uk; 2Department of Renal Surgery, Queen Elizabeth Hospital Birmingham, University Hospitals Birmingham NHS Foundation Trust, Birmingham B15 2TH, UK; Andrew.Ready@uhb.nhs.uk; 3Department of Endocrinology, Queen Elizabeth Hospital Birmingham, University Hospitals Birmingham NHS Foundation Trust, Birmingham B15 2TH, UK; Neil.Gittoes@uhb.nhs.uk; 4Centre for Endocrinology, Diabetes and Metabolism, University of Birmingham, Birmingham Health Partners, Birmingham B15 2TT, UK; 5Faculty of Health and Life Sciences, Aston University Medical School, Birmingham B4 7ET, UK

**Keywords:** coronavirus, COVID-19, parathyroid, hypoparathyroidism, hyperparathyroidism

## Abstract

The coronavirus disease, COVID-19, has caused widespread and sustained disruption to healthcare, not only in the delivery of emergency care, but knock-on consequences have resulted in major delays to the delivery of elective care, including surgery. COVID-19 has accelerated novel pathways for delivering clinical services, many of which have an increased reliance on technology. COVID-19 has impacted care for patients with both hypoparathyroidism and hyperparathyroidism. The role of vitamin D in the prevention of severe COVID-19 infection has also been widely debated. Severe hypocalcemia can be precipitated by infection in patients with hypoparathyroidism. With this in mind, compliance with medical management, including calcium and vitamin D supplementation, is crucial. Technology in the form of text message reminders and smartphone apps may have a key role in ensuring this. Furthermore, clinicians should ensure that patients are educated on the symptoms of hypocalcemia and the steps needing to be taken should these symptoms be experienced. Patients with primary hyperparathyroidism (PHPT) should be educated on the symptoms of hypercalcemia, as well as the importance of remaining adequately hydrated. In addition, patients should be reassured that the postponement of parathyroidectomy is likely to have negligible impact on their condition; for those with symptomatic hypercalcemia, cinacalcet can be considered as an interim measure.

## 1. Introduction

The coronavirus disease, COVID-19, has placed a significant burden upon healthcare systems globally. As part of efforts to treat those most adversely affected by COVID-19, services offered by healthcare systems have had to adapt rapidly. Medical and nursing staff have been redeployed from endocrine care to other specialties such as intensive care, which have been placed under greater strain due to rising numbers of acutely unwell patients. Elective surgical procedures and outpatient appointments have been postponed to accommodate rising emergency admissions [1,2].

The usual management of parathyroid disease during the COVID-19 pandemic has inevitably been disrupted. With the surfacing of new coronavirus variants in the United Kingdom and South Africa [3,4], the likelihood of a return to “business as usual” appears unlikely for the foreseeable future. Therefore, it is crucial to implement lessons learnt from the first waves of the pandemic to optimize patient care going forward. This review synthesizes the latest evidence on the impact of COVID-19 on patients with parathyroid disease as well as highlights pragmatic approaches that clinicians could consider in the management of parathyroid disease during the COVID-19 pandemic. It is important to note that this is a narrative review and largely represents specialist consensus as there are limited high-quality studies in this area at this stage.

## 2. SARS-CoV-2 and Hypoparathyroidism

### 2.1. Are Patients with Hypoparathyroidism More Susceptible to SARS-CoV-2?

The existing evidence reveals no link suggestive of patients with hypoparathyroidism being more likely to contract the SARS-CoV-2 virus or have worse outcomes. However, there are factors associated with hypoparathyroidism, which have been shown to increase the likelihood of infection.

Parathyroid hormone (PTH) plays a key role in the conversion of 25-hydroxyvitamin D to its active form. Individuals with hypoparathyroidism (lacking PTH) have reduced conversion of vitamin D precursors to their active forms; supplementation with active vitamin D is central to the management of the condition [5]. It should be noted that there is evidence of extra-renal hydroxylation of vitamin D. A systematic review and meta-analysis by Martineau et al. of 25 randomized controlled trials (RCTs) concluded that vitamin D supplementation lowered the risk of acute respiratory tract infection [6]. However, previous meta-analyses have reported conflicting results [7,8]. Furthermore, a previous systematic review noted that whilst observational studies generally showed associations between vitamin D status and risk of respiratory tract infections, data from RCTs showed mixed results [9]. More recently, a review by Grant et al. suggested that high-dose vitamin D supplementation may help lower the risk of contracting certain viral respiratory tract infections [10]; however, at present, there is insufficient evidence to support this claim for COVID-19 due to a lack of RCT data in this area. In the UK, a rapid evidence review was performed by the National Institute for Health and Care Excellence (NICE) and the Scientific Advisory Committee on Nutrition (SACN), and a pragmatic approach has been taken where vitamin D supplementation (at a dose of 400 IU/day (10 mcg)) should be considered in the winter months for those at risk of vitamin D deficiency [11]. There are high rates of vitamin D deficiency in the UK population, and such dosing is unlikely to cause harm, although it is likely that this dose would not be sufficient for some patients, particularly those with confirmed vitamin D deficiency on a standard dose induction regimen (such as 3200 IU/day (80 mcg) colecalciferol or equivalent, for 6 weeks) followed by maintenance supplementation (such as colecalciferol 800 to 1000 IU/day (20–25 mcg) or equivalent). In parathyroid disease, it is vital that patients are educated on the importance of medication adherence, especially during the COVID-19 pandemic. Technology can be utilized to ensure compliance, either through telephone or text reminders [12], or alternatively, patients can be directed toward smartphone medication adherence apps. Moreover, clear communication between primary and secondary care clinicians is fundamental to ensuring a continual supply of prescribed medications for patients [12].

### 2.2. How Does SARS-CoV-2 Affect Hypoparathyroidism?

Generally, avoidance of clinically apparent hypocalcemia is a central treatment aim in the management of hypoparathyroidism [12]. Historically, it is known that viral infections can precipitate hypocalcemia [13,14,15]. Singh et al. have highlighted the underlying mechanisms behind severe COVID-19 infection and hypocalcemia [16]. Patients severely affected by COVID-19 have higher levels of unbound [17] and unsaturated fatty acids [18]. The high levels of unsaturated fatty acids can precipitate a cytokine storm [19] and also bind calcium, leading to reduced calcium levels [20]. Furthermore, another study has suggested that lower levels of vitamin D may be associated with more severe COVID infection and therefore a greater likelihood of hypocalcemia [21].

Hypocalcemia has been commonly observed amongst COVID-19 patients. Filippo et al. in their retrospective cohort study of 531 COVID-19 patients admitted to the Emergency Department of San Raffaele Hospital in Milan found that between 78.6 and 82.0% of patients were hypocalcemic, with this being severe in 1.9% of patients [22]. Furthermore, hypocalcemia was found to be a significant predictor of hospitalization with an odds ratio of 4.15 (2.21–7.78, *p* < 0.001). Hypocalcemia was also associated with more severe infection and was predictive of Intensive Care Unit (ICU) admission and death in univariate but not multivariate analyses [22]. This finding of lower calcium levels being associated with more adverse outcomes has been reiterated by two other studies [23,24].

However, it is important to note that this relationship may be more of an association rather than causal. For example, patients who are more ill may present with greater electrolyte derangement such as hypocalcemia and as a result have poorer outcomes, as opposed to hypocalcemia being a specific causal factor for a poorer outcome.

## 3. Advice for Patients with Hypoparathyroidism

Standard therapy for chronic hypoparathyroidism consists of calcium and active vitamin D supplementation (such as alfacalcidol or calcitriol), and some specialists also use “native” vitamin D (such as colecalciferol or ergocalciferol). The goals of chronic management stated in international guidelines include the following: (i) to prevent clinical features of hypocalcemia, (ii) to maintain serum calcium in the low normal range, (iii) to keep the calcium–phosphate levels at target (below 55 mg^2^/dL^2^ which is 4.4 mmol^2^/L^2^), (iv) to avoid hypercalciuria, (v) to avoid hypercalcemia, and (vi) to avoid renal or extra-skeletal calcification [25]. In the case of suspected or confirmed SARS-CoV-2 infection, it is important that patients with hypoparathyroidism continue to comply with vitamin D and calcium supplementation to minimize risk of hypocalcemia. General management principles during the pandemic are listed in Table 1. Patients who take “native” vitamin D should also continue. Patient awareness of symptoms of hypocalcemia, including muscle cramps and paresthesia, is vital so that patients can take additional calcium supplementation (500–1000 mg), in addition to increasing intake of calcium abundant food and drink such as milk. Empirical changes in alfacalcidol are not generally advised and should be guided by the patient’s specialist physician or endocrinologist. Patients should also be made aware of symptoms that may be indicative of hypercalcemia, such as having increased thirst, passing more urine than normal, fatigue, and lethargy. However, if symptoms persist, patients should be aware of the relevant pathways for accessing urgent medical advice [12]. Acute hypocalcemia requiring hospital admission should be managed in line with existing national/international guidelines [25,26]. This can occur as an early complication of anterior neck surgery or in patients who are not compliant with calcium/vitamin D supplementation or as a complication of malabsorption. Symptoms include muscle cramps, paresthesia, and carpopedal spasm. Treatment with intravenous (IV) calcium is indicated in acute symptomatic cases (international guidelines advise the use of 1–2 ampoules of 10% calcium gluconate containing 90–180 mg elemental calcium in 50 mL of 5% dextrose over 10–20 min followed by a slower infusion of calcium gluconate 0.5–1.5 mg/kg/h over 8–10 h) before initiating dose titration of oral calcium supplements and vitamin D analogues [25].

Other recognized symptoms associated with SARS-CoV-2 infection are diarrhea and vomiting [27]. Should patients experience diarrhea and vomiting, they should seek medical advice urgently due to the possibility of poor absorption and need for medication doses to be altered [12].

## 4. Can SARS-CoV-2 Cause Hypoparathyroidism?

Evidence for this is lacking; however, an interesting case was reported by Elkattawy et al., where SARS-CoV-2 infection as the cause for hypoparathyroidism was suspected. The authors report the case of a 46-year-old male patient with an unremarkable medical history, following admission for hypoxic respiratory failure secondary to SARS-CoV-2 infection. Other well-established causes of hypoparathyroidism such as genetic causes or malignancy were excluded [28]. Given that this is a case report, limited conclusions can be drawn from it; however, it does suggest the need for further investigation in this field to identify if there is any link between SARS-CoV-2 infection and parathyroid function.

## 5. SARS-CoV-2 and Primary Hyperparathyroidism (PHPT)

Primary hyperparathyroidism (PHPT) is a common condition characterized by inappropriately raised PTH secretion, which may result in hypercalcemia and associated symptoms [29]. PHPT has a slow onset, with many patients being unaware of the presence of the condition until symptoms of hypercalcemia develop. Due to the slow-evolving nature of the condition, diagnostic investigations for localizing the disease and determining the end-organ impact of the disease, such as on bone mineral density and renal function (including nephrolithiasis) should be delayed if necessary regarding the COVID-19 pandemic context [12].

Patients with PHPT should be reminded of the symptoms of hypercalcemia such as vomiting, nausea, and abdominal pain. This can be achieved by directing patients to reliable sources of online information such as the UK National Health Service (NHS) website, where information is displayed in a simple and understandable format. In addition, patients should be informed about the importance of staying hydrated, given that dehydration can precipitate hypercalcemia [12]. SARS-CoV-2 infection is associated with symptoms such as a fever, diarrhea, and vomiting [25], all contributing to dehydration. Therefore, it is vital that patients remain well-hydrated should such symptoms develop. Failure to achieve adequate hydration may amount in a deterioration of their condition and progressive hypercalcemia, which may demand emergency admission [12]. A summary of key priorities for managing PHPT during the pandemic is provided in Table 2. 

Some international guidelines and consensus statements [30,31] that were written prior to the emergence of COVID-19 outline the following indications for surgery in asymptomatic patients for the treatment of PHPT: (1) age < 50 years; (2) serum calcium > 1 mg/dL or > 0.25 mmol/L of the upper limit of the reference range; (3) Bone Mineral Density (BMD) T-score of </= −2.5 at lumbar spine, femoral neck, or distal third of the radius or evidence of a low energy fracture; (4) glomerular filtration rate (GFR) of < 60 mL/min. Symptomatic disease is also accepted as an indication for surgery. Medical management options include vitamin D supplementation to treat and prevent deficiency/insufficiency, bisphosphonates where there is evidence of reduced BMD, and cinacalcet, which can be effective in lowering serum calcium and can be considered for symptomatic PHPT where surgery is not an option. It is important to note that data included in these guidelines had limited follow up for medical therapies and is “insufficient to justify medical therapy as an alternative to surgery” [30,31].

Elective parathyroidectomy for PHPT is mostly postponed during the peaks of the pandemic due to a shortage of both staff and physical resources as well as risk associated with entering healthcare systems. If we take our center (which has had a high local prevalence of COVID-19, ≈12,500 cases in 11 months) as an example, we observed a 62% reduction in completed parathyroidectomies in 2020 compared to the previous year (*n* = 31 in 2020 vs. 81 in 2019). Nine of the 31 cases carried out in 2020 were done pre-pandemic, so 22 cases were carried out in the months of March to December. Based on case numbers from the previous 48 months, a mean of seven cases per month were carried out, which is indicative of a 69% reduction in expected operating activity since the start of the pandemic. With the impact of the “second wave” of infections, operating has been halted in recent months, so when taking the 14-month period from the start of 2020 to the end of February 2020, 31 parathyroidectomies were completed when we could have expected to have carried out 98 at previous activity levels. This equates to an absolute total of 67 cases that were not carried out. This has impacted on waiting lists, which include some 73 patients. The majority of these have breached the 18 weeks to treatment threshold, which is a key metric in the UK NHS system. It is clear that there will be ongoing delays for patients to access definitive management. For example, in our service, it would take 10 months to clear this backlog at previous rates of activity. We have also experienced a reduction in referrals during the pandemic with only 55 referrals entering the system in 2020, which is a reduction of 25–30 patients compared to previous years. We would expect an increase in referrals as routine primary care and secondary care activity increases with internal projections, indicating that a further 105 patients will be added to our waiting list for parathyroidectomy this year and would require a doubling of operating activity to clear the waiting list in a year. Overall, this represents a considerable challenge, particularly since the indication for surgery is for long-term health benefits as opposed to for acute life-threatening conditions or for cancers regarding which services will have to compete with for theater space.

Clinicians should inform patients who had planned to undergo parathyroidectomy during this time period that the clinical impact of this postponement is likely to be minimal due to the slow-evolving nature of the condition. These decisions are complex and depend upon patient-specific factors as well as the wider context of the pandemic at the time and how this has impacted on access to services. In special circumstances, a multi-disciplinary team approach should be adopted for example in the case of hyperparathyroidism in pregnancy, suspected parathyroid carcinoma, or where the condition is impacting on other co-morbidities; thankfully, however, these circumstances are rare. For patients experiencing symptomatic hypercalcemia, medical management with cinacalcet may be considered until elective parathyroidectomy can be performed [12]. However, this treatment does not have approval or market authorization for use in PHPT in many countries. It is especially important that patients keep well hydrated after cinacalcet intake in view of the impact of the drug on urinary calcium excretion via the activation of Calcium Sensing Receptor (CaSR) expressed in the thick ascending limb of the loop of Henle.

## 6. General Challenges Associated with Delivering Outpatient Parathyroid Disease Care

### 6.1. Greater Telemedicine Use

Due to the COVID-19 pandemic, numerous restrictions have been placed on outpatient care. Namely, there has been a reduction in the number of face-to-face consultations, which have been substituted with telephone or video consultations. Whilst this was initially unfamiliar territory for both clinicians and patients, in most instances, this has posed no obstacle to the provision of high-quality parathyroid disease care. Patients should continue to be well-informed regarding their care by being sent routine clinic letters from clinicians [12]. Within other realms of endocrinology such as in diabetes care, telemedicine has been shown to yield greater satisfaction amongst both patients and clinicians [32,33]. Whilst there are no studies examining the cost-effectiveness of telemedicine compared to face-to-face consultations in the context of parathyroid disease care, studies for diabetic care have found telemedicine to be cost-effective [34]. Given the increasing financial burden on healthcare systems, it is likely that beyond the COVID-19 pandemic, a high proportion of parathyroid disease care consultations will remain as telemedicine consultations in some healthcare settings.

### 6.2. Reduced Access to Blood Tests and Urine Tests

Another key aspect of parathyroid disease care is monitoring bloods and urine tests. This is vital to ensure that disease is adequately controlled and informs decisions regarding the patient’s treatment plan [12]. However, due to greater restrictions, there is reduced access to these tests. Due to this limited access, clinicians should ensure blood and urine tests are necessary and have a role in possibly adapting the management plan for a patient. Furthermore, tests should be conducted at areas outside of the hospital setting, such as at GP surgeries, or the “drive-through” phlebotomy service model could be developed further [12].

## 7. Conclusions

COVID-19 has had a significant impact on parathyroid disease care, mainly due to delays in elective surgery and increased restrictions on outpatient care leading to fewer face-to-face consultations. Key priorities for the management of hypoparathyroidism include ensuring medication adherence and promoting patient education regarding the symptoms of hypocalcemia. Furthermore, if patients experience diarrhea or vomiting, patients should seek medical advice, as alterations to medication doses are likely to be required. With regard to hyperparathyroidism, patients should be educated on the symptoms of hypercalcemia and the importance of remaining well-hydrated. Despite elective parathyroidectomies being widely postponed, patients should be reassured that this will have little adverse impact to their condition. Given the unpredictable nature of the COVID-19 pandemic, endocrinologists should be aware of these priorities and ensure that they are implemented in the care of parathyroid disease patients.

## Figures and Tables

**Table 1 jcm-10-00920-t001:** A summary of the key priorities for managing hypoparathyroidism during the coronavirus disease 2019 (COVID-19) pandemic [12].

Patient education on the symptoms of hypocalcemia and hypercalcemia. Paresthesia and muscle cramps may be indicative of hypocalcemia. Increased thirst, polyuria, nocturia, and lethargy may indicate that over-treatment has resulted in hypercalcemia. Patients should be made aware of appropriate next steps should symptoms develop
Emphasizing the importance of adherence to prescription medication to patients (utilizing text message reminders or encouraging the use of medication adherence smartphone applications)
Informing patients of the need to seek medical advice should diarrhea or vomiting develop, as medication doses may need to be altered.

**Table 2 jcm-10-00920-t002:** A summary of the key priorities for managing primary hyperparathyroidism (PHPT) during the COVID-19 pandemic [12].

Patient education on the symptoms of hypercalcemia such as abdominal pain, nausea, and vomiting and informing patients of the appropriate next steps should symptoms of hypercalcemia develop
Emphasizing the importance of remaining adequately hydrated
Explain to patients who have had their parathyroidectomy postponed that their condition is unlikely to deteriorate, as PHPT is a slow-evolving condition
Consider cinacalcet to manage symptomatic hypercalcemia

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
