# Peer review of "Management of Parathyroid Disease during the COVID-19 Pandemic"

_jcm, 2021, doi:10.3390/jcm10050920_

Round 1

Reviewer 1 Report

The authors presented an “overview” about different parathyroid diseases during Covid 19 pandemic. In summary, the manuscript does not fulfill the criteria of a valuable review.  Additionally, there are no helpful advices for the management, because all the advices that have been given, are standard procedures for example in case of symptomatic hyperparathyroidism. The manuscript should be thoroughly revised again.

Author Response

Thank you for taking the time to review this paper. 

We thought that it was important to make a review of the current literature in this area and provide some pragmatic advice available for a general internal medicine audience who will not be as familiar with the management changes that we have carried out in speciality. It is a common scenario, of great topicality and importance. Pragmatic advice on the impact of the infection on parathyroid diseases is much needed and from our discussions with patients and general internal medicine colleagues would be welcomed.

What's more, we have made some changes in line the the reviewers comments.

Reviewer 2 Report

congretulation to this nice paper 

Author Response

Many thanks for taking the time to review our paper - we have made some further revisions in line with the advice of all of the reviewers. 

Best wishes

Reviewer 3 Report

This review is consensual and well written and should be useful for the care of patients with parathyroid diseases during the COVID-19 pandemic. I have few comments: - Regarding this point : "pragmatic approach has been taken where vitamin D supplementation (at a dose of 400 IU/day (10mcg)) should be considered in the winter months, for those at risk of vitamin D deficiency [11]. There are high rates of vitamin D deficiency in the UK population and such dosing is unlikely to cause harm...". This is also likely that such a low dose will not be able to correct vitamin D insufficiency and will therefore be of little help to potentially reduce the risk of COVID-19 infection and/or complications. This point should be highlighted. - In the first paragraph "Are patients with hypoparathyroidism more susceptible to SARS-CoV-2?", it is unclear wether patients with hypoparathyroidism should be aware of the importance of taking active vitamin D, native vitamin D or both?. Of note, native vitamin D should also be useful because it can be converted into calcitriol independently of PTH outside the kidney (including in cells of the immune system). - "This finding of lower calcium levels being associated with more adverse outcomes has been reiterated by a number of other studies [23,24]." Please replace "by a number of other" by " two other studies" - "Due to the slow-evolving nature of the condition, diagnostic investigations for localising the disease and determining the end-organ impact of the disease, such as on bone mineral density and renal function should be delayed [12].". Please add nephrolithiasis in this sentence. Please add after "should be delayed": "if necessary regarding the COVID-19 pandemic context" - "For patients experiencing symptomatic hypercalcaemia, medical management with cinacalcet may be considered until elective parathyroidectomy can be performed [12].". Please add that hyperhydration is especially important after cinacalcet intake due to the rise of urinary calcium excretion after cinacalcet intake (secondary to the action of cinacalcet on the CaSR expressed in the thick ascending limb of Henle). - "Reduced access to blood tests". Please change for "Reduced access to blood and urinary tests". Indeed, urinary calcium excretion monitoring is often mandatory (ie. to ensure the absence of iatrogenic high urinary calcium excretion during the treatment of hypoparathyroidism".

Author Response

Thank you for your expertise and for taking the time to review our manuscript. We found your comments to be very helpful and think that your insights have improved the paper further. I have made amendments to the manuscript in line with your comments - as summarised below: 

This review is consensual and well written and should be useful for the care of patients with parathyroid diseases during the COVID-19 pandemic.

I have few comments: - Regarding this point : "pragmatic approach has been taken where vitamin D supplementation (at a dose of 400 IU/day (10mcg)) should be considered in the winter months, for those at risk of vitamin D deficiency [11]. There are high rates of vitamin D deficiency in the UK population and such dosing is unlikely to cause harm...". This is also likely that such a low dose will not be able to correct vitamin D insufficiency and will therefore be of little help to potentially reduce the risk of COVID-19 infection and/or complications. This point should be highlighted.

Many thanks for this comment – yes we agree and have had similar discussions in our practice and more widely in relation to this. We have amended the manuscript as follows:

“There are high rates of vitamin D deficiency in the UK population and such dosing is unlikely to cause harm, although it is likely that this dose would not be sufficient for some patients, particularly those with confirmed vitamin D deficiency where a standard dose induction regimen (such as 3200IU/day colecalciferol (25mcg) or equivalent, for 6 weeks) followed by maintenance supplementation (such as colecalciferol 800 to 1000IU/day or equivalent).”

- In the first paragraph "Are patients with hypoparathyroidism more susceptible to SARS-CoV-2?", it is unclear wether patients with hypoparathyroidism should be aware of the importance of taking active vitamin D, native vitamin D or both?. Of note, native vitamin D should also be useful because it can be converted into calcitriol independently of PTH outside the kidney (including in cells of the immune system).

Thank you for this comment, I have made the following edits:

This sentence has been included in the advice section:

Patients who take ‘native’ vitamin D such as colecalciferol in such cases this should also continue.

Also I have changed the sentence in the ‘Are patients with hypoparathyroidism more susceptible to SARS-CoV-2?’ section as follows:

‘Individuals with hypoparathyroidism (lacking PTH), have reduced conversion vitamin D precursors to their active forms. and ‘It should be noted that there is evidence of extra-renal hydroxylation of native vitamin D.’

It previously read that they are unable to convert…

- "This finding of lower calcium levels being associated with more adverse outcomes has been reiterated by a number of other studies [23,24]." Please replace "by a number of other" by " two other studies"

Many thanks I have made this change.

- "Due to the slow-evolving nature of the condition, diagnostic investigations for localising the disease and determining the end-organ impact of the disease, such as on bone mineral density and renal function should be delayed [12].". Please add nephrolithiasis in this sentence. Please add after "should be delayed": "if necessary regarding the COVID-19 pandemic context"

Many thanks I have made this change and I agree that it is helpful to add the caveat that it only needs to be delayed if necessary.

- "For patients experiencing symptomatic hypercalcaemia, medical management with cinacalcet may be considered until elective parathyroidectomy can be performed [12].". Please add that hyperhydration is especially important after cinacalcet intake due to the rise of urinary calcium excretion after cinacalcet intake (secondary to the action of cinacalcet on the CaSR expressed in the thick ascending limb of Henle).

Many thanks for this important detail – I have amended the manuscript to reflect this.

- "Reduced access to blood tests". Please change for "Reduced access to blood and urinary tests". Indeed, urinary calcium excretion monitoring is often mandatory (ie. to ensure the absence of iatrogenic high urinary calcium excretion during the treatment of hypoparathyroidism".

Thank you, I have made this change.

Reviewer 4 Report

The manuscript is well-written and exhaustive. I have just  a comment about the parathyroidectomy: is it always safe to delay it or there are clinical situation in which the physician should pay particular attention?

Author Response

Thank you for giving your time and expertise for this review. I think that this is an excellent point and can think of some scenarios in our practice where this would need some more thought. I have included the following in the manuscript to reflect this:

‘These decisions are complex and depend upon patient specific factors as well as the wider context of the pandemic at the time and how this has impacted on access to services. In special circumstances a multi-disciplinary team approach should be adopted for example in the case of pregnancy, suspected parathyroid carcinoma or where the condition is impacting on other co-morbidities, thankfully however these circumstances are rare.’

Round 2

Reviewer 1 Report

Thank you very much for your explanations!

Author Response

Many thanks for taking the time to review our manuscript. 

I have added a full response to the editors which can be included here.